# Phenolic Compounds from *Sonchus arvensis* Linn. and *Hemerocallis citrina* Baroni. Inhibit Sucrose and Stearic Acid Induced Damage in *Caenorhabditis elegans*

**DOI:** 10.3390/molecules28041707

**Published:** 2023-02-10

**Authors:** Qin An, Lei Zhang, Xiyue Qin, Xiong Wang, Wenli Wang, Qingyong Meng, Yali Zhang

**Affiliations:** 1College of Food Science and Nutritional Engineering, China Agricultural University, Beijing 100083, China; 2College of Biological Sciences, China Agricultural University, Beijing 100193, China

**Keywords:** Sonchus arvensis Linn., Hemerocallis citrina Baroni., phenolic compounds, Caenorhabditis elegans, antioxidant

## Abstract

*Sonchus arvensis* Linn. and *Hemerocallis citrina* Baroni. have been reported to improve body resistance. However, the underlying mechanism is not clear. In this study, *Sonchus arvensis* Linn. phenolic compounds (SAP) and *Hemerocallis citrina* Baroni. phenolic compounds (HCP) were extracted and their protective effects in *Caenorhabditis elegans* evaluated. SAP and HCP showed considerably different phenolic compositions. In the normal *C. elegans* model, HCP exhibited better effects in promoting growth than SAP. In the sucrose-incubated *C. elegans* model, both SAP and HCP showed positive effects against the high-sucrose-induced damage. In the stearic acid-incubated *C. elegans* model, both SAP and HCP improved lifespan, reproductive ability and growth, while HCP had a more evident effect than SAP on reproductive ability. The TGF-β signaling pathway was confirmed to be involved in the protective effects of SAP and HCP. The antioxidant ability of SAP was also found to be related to *skn-1*. Our study shows that both SAP and HCP have protective effects against high sucrose- or high stearic acid-induced damage.

## 1. Introduction

Carbohydrates and fats are the main sources of organic materials used for supporting the human body. A reasonable intake of carbohydrates and fats has a positive impact on human activities; however excessive intake may prove harmful to the human body, leading to the occurrence of metabolic syndrome. With an improvement in the living standards of people, dietary structure gradually shifts towards the direction of high sugar and fat consumption. Long-term intake of a diet high in sugar and fat leads to abnormal blood pressure and blood lipid levels, cognitive dysfunction, and damage to male reproductive ability. It may also make offspring vulnerable to chronic diseases, such as type 1 diabetes and reproductive disorders [1,2]. Owing to the complex background mechanism of metabolic syndrome caused by high-sugar and high-fat diets, it has become very difficult to develop effective drugs to prevent or improve metabolic syndrome. Currently, common drugs in the market are targeted to intervene in a single group of patients with metabolic syndrome. However, the disadvantage of drug therapy lies in the use of a single target, which can easily cause side effects [3]. Accordingly, the search for natural compounds as therapeutic agents that can effectively intervene in metabolic syndrome has become a research hotspot.

*Sonchus arvensis* Linn. is a traditional food in northern China. The extract of *Sonchus arvensis* Linn. has many pharmacological effects, including antioxidant, hepatoprotective, kidney-protective, antibacterial, and anti-fatigue properties [4,5,6]. The extracts of *Sonchus arvensis* Linn. scavenge the free radicals, which may be due to the presence of functional components include rutin, quercetin, catechin, and myricetin [7]. Previous studies have shown that *Sonchus arvensis* Linn. phenolic compounds (SAP) have considerable antioxidant capabilities [8]. However, its mechanism of action requires further investigation.

*Hemerocallis citrina* Baroni., commonly known as Citron daylily and long yellow daylily, is consumed as a vegetable in China and other Asian countries [9]. It is rich in multiple compounds, such as carbohydrates, protein, lipid, and vitamins. It also contains many volatile substances, including phenols, alcohols, and acids. *Hemerocallis citrina* Baroni. exerts various bioactivities including anti-gloom, anti-tumor, antioxidant, cholesterol-lowering, and anti-aging activities [10,11,12]. The strong antioxidant activity of *Hemerocallis citrina* Baroni. can be attributed to the action of antioxidant compounds such as phenolic compounds [13]. *Hemerocallis citrina* Baroni. phenolic compounds (HCP) have been shown to play many physiological functions, such as enhancing stress resistance and hepatoprotection [14,15]. HCP has effective protective properties, but the underlying mechanism is not clear.

*Caenorhabditis elegans* is an ideal model for studying lipid and sugar resistance because of its low cost, short lifespan, rapid reproductions cycle, and ease of observation [16,17]. In addition, the whole genome sequence of *C. elegans* is known, and more than 60% of the genes are homologous to human disease genes [18]. In this study, we employed a *C. elegans* model to investigate the protective effects of SAP and HCP in vivo.

In *C. elegans*, transforming growth factor-β (TGF-β) superfamily ligands participate in cell identification, growth, and development. *Dbl-1* and *daf-7* are the ligands belong to TGF-β superfamily and their function has been explained more clearly [19]. The *C. elegans* insulin signaling pathway links energy metabolism with growth, development, reproductive, longevity, and behavior [20]. This insulin signaling pathway is regulated by insulin-like peptide ligands that bind to the insulin/IGF-1 transmembrane receptor ortholog *daf-2* [21]. In *C. elegans*, *skn-1*, the downstream regulator of *daf-2*, is required for both oxidative stress resistance and anti-aging through its accumulation in the intestinal nuclei to promote the detoxication of target genes [22].

In the present study, to better understand the bioactivity of SAP and HCP, the phenolic compounds in the SAP and HCP were analyzed using ultra-performance liquid chromatography coupled to triple-quadrupole tandem mass spectrometer (UPLC-QQQ-MS/MS). In addition, we evaluated the effects of SAP and HCP in normal *C. elegans*. Furthermore, the role of SAP and HCP in protecting *C. elegans* against sucrose or stearic acid damage was explored. The results will help investigate the mechanism of SAP and HCP and define health claims related to the consumption of *Sonchus arvensis* Linn. and *Hemerocallis citrina* Baroni.

## 2. Results

### 2.1. Extraction and Component Analysis of SAP and HCP

The yields of SAP and HCP were 23% and 39%, respectively. We used a method in the MRM mode of UPLC-QQ-MS/MS system for qualitative and quantitative analysis of phenolic compounds in SAP and HCP. The *Sonchus arvensis* Linn. is rich in rutin, quercetin, chlorogenic acid and other phenolic compounds [23]. Previous study showed that there are 25 polyphenols in *Hemerocallis citrina* Baroni [24].

As shown in Figure 1 and Figure 2, seven phenolic compounds were detected in SAP and eight phenolic compounds were detected in HCP. The phenolic compositions of SAP and HCP are summarized in Table 1. The phenolic compounds found in both SAP and HCP were rutin, chlorogenic acid, and quercetin, although their contents were extremely different. The major phenolic compounds found in SAP and HCP were chlorogenic acid and rutin, respectively. These findings indicate that SAP and HCP, with different phenolic compositions, may have different physiological activities.

### 2.2. Effects of SAP and HCP on the Body Length, Progeny Production and Lifespan of C. elegans

Based on the results of the preliminary experiment, *C. elegans* were treated with SAP at concentrations ranging from 0 to 2000 μg/mL and HCP at concentrations ranging from 0 to 1200 μg/mL. As shown in Figure 3A and Table 2, the survival curve of *C. elegans* treated with 500 and 1000 μg/mL SAP were obviously shifted to the right. In addition, treatment with 500 and 1000 μg/mL SAP significantly (*p* < 0.05) prolonged the average lifespan of *C. elegans*. As shown in Figure 3B and Table 2, the survival curve of 1200 μg/mL HCP was shifted to the left. Treatment with HCP at concentrations ranging from 400 to 1200 μg/mL had no significant effect on the average lifespan of *C. elegans*. These results indicated that between the two types of extracts, SAP exhibited a more positive effect that prolonged the lifespan of *C. elegans*.

As shown in Figure 3C,D, after treatment with SAP or HCP, the average number of eggs laid by *C. elegans* on each day initially increased, and then decreased. *C. elegans* treated with control and different concentrations of SAP and HCP entered the spawning period on the second day and ended spawning on the sixth day. As shown in Figure 3E, after treatment with 500, 1000, and 2000 μg/mL SAP, the total number of eggs in the entire spawning period significantly (*p* < 0.05) increased, especially with 500 μg/mL treatments, showing the most statistically different effect among the three concentrations of SAP groups. As shown in Figure 3F, after treatment with 400 μg/mL HCP, the total number of eggs laid by *C. elegans* in the entire spawning period decreased, whereas treatment with 800 and 1200 μg/mL HCP significantly (*p* < 0.05) increased the number of eggs, and 1200 μg/mL HCP showed the most statistically different effect, increasing by 51%. Taken together, SAP and HCP improved the reproductive ability of *C. elegans*, especially low concentration SAP and high concentration HCP.

As shown in Figure 3G, compared with that in the control group, except for treatment with SAP at 1000 μg/mL, there was no significant difference in the length of *C. elegans* after treatment with the other concentrations of SAP. In the growing phase, the body length of *C. elegans* grew rapidly and reached a maximum of 1.2 mm on the fifth day, after treatment with SAP at 500 and 1000 μg/mL. However, treatment with 2000 μg/mL SAP brought forward the appearance of *C. elegans* aging. As shown in Figure 3H, as *C. elegans* entered the spawning period, HCP treatment rapidly promoted the growth of *C. elegans*, especially at a concentration of 1200 μg/mL. In addition, 800 and 1200 μg/mL HCP treatment significantly (*p* < 0.05) increased the maximum length of *C. elegans*, which reached a maximum length of 1.4 mm on the fifth day.

### 2.3. Effects of SAP and HCP on the Body Length, Progeny Production and Lifespan of Sucrose-incubated C. elegans

As shown in Table 3, compared to that of the control group, the average lifespan of *C. elegans* treated with 100 mM sucrose was significantly (*p* < 0.05) decreased. The average lifespan of sucrose-incubated *C. elegans* treated with SAP and HCP significantly (*p* < 0.05) increased compared with that of the control group. As shown in Figure 4A,B, compared to those of the control group, the survival curves of the HCP and SAP groups were considerably shifted to the right before the tenth day. Moreover, throughout the observation period, the survival curves of the HCP and SAP groups were obviously shifted to the right compared to those of the sucrose-incubated model group. These results indicated that the lifespan of sucrose-incubated *C. elegans* could be recovered by SAP and HCP treatment.

As shown in Figure 4C,D, after treatment with 100 mM sucrose, the spawning periods were delayed by one day. A delay in the spawning period indicates inhibition of *C. elegans* development. As shown in Figure 4E,F, compared with the control group, the total number of eggs laid by *C. elegans* treated with sucrose significantly (*p* < 0.05) decreased. After being treated with SAP, the total number of eggs laid by sucrose-incubated *C. elegans* increased with increasing SAP concentrations. However, HCP treatment increased the total number of eggs laid by sucrose-incubated *C. elegans* without an obvious dose-response. Taken together, 100 mM sucrose caused significant damage to the reproductive ability of *C. elegans*, and the damage decreased with SAP and HCP treatment.

As shown in Figure 4G,H, compared with the body length in the control group, 100 mM sucrose significantly (*p* < 0.05) inhibited the growth of *C. elegans* but had no significant effect on the maximum body length of *C. elegans*. In the growing phase, treatment with 500 to 2000 μg/mL SAP significantly (*p* < 0.05) increased the body length of the sucrose-incubated *C. elegans*. In addition, after SAP treatment, the body length of sucrose-incubated *C. elegans* reached a maximum length of 1.2 mm on the fifth day, which was 1.1 times the length of the *C. elegans* in the model group. However, there was no significant difference in the length of sucrose-incubated *C. elegans* after HCP treatment.

### 2.4. Effects of SAP and HCP on the Body Length, Progeny Production and Lifespan of Stearic Acid-Incubated C. elegans

As shown in Figure 5 and Table 4, compared to the control group, 352 mM stearic acid treatment reduced the average lifespan of *C. elegans* by 2%. The average lifespan of stearic acid-incubated *C. elegans* treated with SAP or HCP initially increased and subsequently decreased. Treatment with 500–1000 μg/mL SAP significantly (*p* < 0.05) prolonged the average lifespan of stearic acid-incubated *C. elegans*, whereas treatment with 2000 μg/mL SAP had a weak effect on that of stearic acid-incubated *C. elegans*. In contrast, treatment with 400 μg/mL HCP had a weak effect on the average lifespan of stearic acid-incubated *C. elegans*, whereas 800–1200 μg/mL HCP significantly (*p* < 0.05) prolonged the lifespan of stearic acid-incubated *C. elegans*. This indicates that SAP or HCP treatment could improve the average lifespan of stearic acid-incubated *C. elegans*, although the effective concentration of the two extracts was different.

As shown in Figure 5C–F, after treatment with 352 mM stearic acid, the spawning periods were delayed by one day, and the total number of eggs laid by *C. elegans* significantly (*p* < 0.05) decreased. As shown in Figure 5E, after treatment with 500 and 1000 μg/mL SAP, the total number of eggs laid by stearic acid-incubated *C. elegans* significantly (*p* < 0.05) increased, especially with 500 μg/mL SAP treatment. However, 2000 μg/mL SAP had no significant effect on the reproductive ability of stearic acid-incubated *C. elegans*. This result was similar to the lifespan of stearic acid-incubated *C. elegans*, indicating that when SAP was added at a concentration of 2000 μg/mL, its protective effect was weakened. As shown in Figure 5F, after treatment with 400, 800, or 1200 μg/mL HCP, the total number of eggs laid by stearic acid-incubated *C. elegans* significantly (*p* < 0.05) increased, and 1200 μg/mL HCP had the most positive effect. Taken together, the intake of 352 mM stearic acid caused significant damage to the reproductive ability of *C. elegans*, and the damage decreased after SAP and HCP treatment.

As shown in Figure 5G,H, treatment with 352 mM stearic acid significantly (*p* < 0.05) inhibited the growth of *C. elegans*, and reduced the maximum length by 20% on the fifth day compared with that in the control group. As shown in Figure 5G, the body length of the stearic acid-incubated *C. elegans* treated with 500–2000 μg/mL SAP was similar to that in the control group. In other words, SAP treatment significantly (*p* < 0.05) promoted the growth of stearic acid-incubated *C. elegans*, especially at 500 μg/mL. As shown in Figure 5H, treatment with 400–1200 μg/mL HCP significantly (*p* < 0.05) increased the body length of stearic acid-incubated *C. elegans* on the second day. In addition, 1200 μg/mL HCP had a positive effect on the length of stearic acid-incubated *C. elegans* on the third day. However, compared with that in the control group, 1200 μg/mL HCP treatment shortened the body length of *C. elegans* on the fifth and sixth days.

### 2.5. Effects of SAP and HCP on Gene Expressions in C. elegans

To explore the protective mechanisms of SAP and HCP, we detected the mRNA expression level of related genes (Table 5). The worms were treated with SAP (500 μg/mL) or HCP (1200 μg/mL).

As shown in Figure 6A, no significant difference was observedin the expression of *skn-1* in normal *C. elegans*. It was found that 100 mM sucrose and 100 μg/L stearic acid had weak effects on the expression of *skn-1*. In sucrose-incubated *C. elegans*, no significant difference was observed in the expression of *skn-1*. In stearic acid-incubated *C. elegans*, the expression of *skn-1* was significantly (*p* < 0.05) increased with SAP (500 μg/mL) treatment, whereas no significant difference was observed with HCP (1200 μg/mL) treatment. As shown in Figure 6B, no significant differences were observed in the expression of *daf-7* in normal and sucrose-damaged *C. elegans*. However, both SAP (500 μg/mL) and HCP (1200 μg/mL) increase the expression of *daf-7* in stearic acid-incubated *C. elegans*, and there was no significant difference between the effects of the two treatments. As shown in Figure 6C, SAP (500 μg/mL) treatment or HCP (1200 μg/mL) treatment had a weak effect on the expression of *dbl-1* in the normal *C. elegans*. Treatment with 100 mM sucrose or 352 mM stearic acid significantly (*p* < 0.05) decreased the expression of *dbl-1*. In sucrose-incubated *C. elegans*, the expression of *dbl-1* was significantly (*p* < 0.05) increased after HCP (1200 μg/mL) treatment, whereas no significant difference was observed with SAP (500 μg/mL) treatment. However, no significant difference was observed in the expression of *dbl-1* in stearic acid-incubated *C. elegans*. As shown in Figure 6D, no significant difference was observed in the expression of *sma-4* in normal *C. elegans*. Treatment with 100 mM sucrose or 352 mM stearic acid significantly (*p* < 0.05) decreased the expression of *dbl-1*. In sucrose-incubated *C. elegans*, the expression of *sma-4* was significantly (*p* < 0.05) increased with HCP (1200 μg/mL) treatment, whereas no significant difference was observed with SAP (500 μg/mL) treatment. In stearic acid-incubated *C. elegans*, the expression of *sma-4* was significantly (*p* < 0.05) decreased with HCP (1200 μg/mL) treatment, while no significant difference was observed with SAP (500 μg/mL) treatment.

## 3. Materials and Methods

### 3.1. Reagents

Chlorogenic acid, hyperin, rutin, apigenin, quercetin, scopoletin, chrysoeriol, caffeic acid, protocatechin, isorhamnetin, taxifolin, and gentisic acid were purchased from Macklin (Shanghai, China). Methyl alcohol, formic acid, and n-Hexane were purchased from Merck Millipore. The sucrose and stearic acid were purchased from Sigma (St. Louis, MO, USA). Na_2_HPO_4_, KH_2_PO_4_, KPO_4_, NaCl, CaCl_2_, MgSO_4_, and NH_4_Cl were purchased from Beijing Lanyi Chemical Products Co. Ltd. (Beijing, China). Sucrose, stearic acid, agar, peptone, cholesterol, and DMSO were purchased from Lablead (Beijing, China). 

### 3.2. Preparation of Phenolic Compunds from Sonchus arvensis Linn. or Hemerocallis citrina Baroni

The leaves of *Sonchus arvensis* Linn. were collected from Xinghua City, Jiangsu Province in China. The buds of *Hemerocallis citrina* Baroni. were collected from Datong City, Shanxi Province in China. The powder of *Sonchus arvensis* Linn. or *Hemerocallis citrina* Baroni. (0.2 g) was extracted using 80% methyl alcohol (5 mL) in an ultrasonic bath for 20 min (SCQ-3201, Shanghai Shengyan Ultrasonic Co., Ltd., Shanghai, China), followed by centrifugation at 7000 rpm and 4 °C (Sigma-Aldrich, Shanghai, China). The supernatant was collected and the residue was re-extracted two more times under the same conditions. The supernatant was diluted to 25 mL and mixed with n-Hexane at a 1:1 ratio. After the mixed liquid was stratified, the upper liquid was removed and lyophilized to yield partially purified *Sonchus arvensis* Linn. phenolic compounds (SAP) or *Hemerocallis citrina* Baroni. phenolic compounds (HCP) by using a vacuum freeze-dryer (Christ, Osterode, Germany). The extraction yield (%) of SAP and HCP was calculated using the following equation: extraction yield (%, *w*/*w*) = (weight of dried SAP or HCP (g)/weight of dry materials (g)) × 100%.

### 3.3. Phenolic Compositions Analysis of SAP and HCP

Phenolic compositions were analyzed using the UPLC-QQQ-MS/MS method. Samples were separated using ACQUITY UPLC BEH C18 (100 mm × 2.1 mm, 1.7 μm, Waters, Milford, MA, USA). The column temperature was set at 30 °C and the sample injection volume was 5 μL. Mobile phase A: methanol and mobile phase B: 0.1% formic acid in 5 mmol/L aqueous acetate were utilized with a gradient elution as follows: 0–1 min, 10% A and 90% B; 1–9 min, 10% A and 90% B; 9–11 min, 90% A and 10% B; 11–11.1 min, 100% A; 11.1–13 min, 10% A and 90% B. Post run lasted for 2 min to balance and wash the column. The flow rate was 0.25 mL/min, and eluate was monitored with a DAD detector (Waters, Milford, MA, USA) at 280 nm. Mass spectrometry data were obtained simultaneously using an electrospray ionization source in the negative ionization mode (Waters, Milford, MA, USA). The optimal source conditions were a drying gas temperature of 350 °C, nebulizer pressure of 650 L/h, and a capillary voltage of 3000 V. Phenolic compounds standards were used to identify and quantify the corresponding peaks.

### 3.4. C. elegans Culture and Treatment

Wild-type N2 strains were obtained from the Caenorhabditis Genetics Center (University of MN, Minneapolis, MN, USA) and maintained on nematode growth medium (NGM) plates (1.7% agar, 2.5 g/L peptone, 51 mM NaCl, 25 mM KPO_4_ buffer pH 6.0, 5 μg/L cholesterol, 1 mM CaCl_2_, 1 mM MgSO_4_) with *Escherichia coli* OP50 (Rutgers University, New Brunswick, NJ, USA) as food resource, at 20 °C. To obtain a synchronized population of worms, the eggs were collected in a certain amount of time. M9 buffer (41 mM Na_2_HPO_4_, 15 mM KH_2_PO_4_, 8.6 mM NaCl, 19 mM NH_4_Cl) was used to wash the eggs. Sucrose plates were prepared by adding sucrose (100 mM, sterile filtered) into NGM [28]. Stearic acid plates were prepared by adding stearic acid (352 mM, sterile filtered) into NGM [28]. SAP or HCP completely dissolved in DMSO and configured to 2 mg/mL stock solution. For assays, SAP (0, 500, 1000, and 2000 μg/mL) or HCP (0, 400, 800, and 1200 μg/mL) were added to *Escherichia coli* OP50 at a rate of 2%.

### 3.5. Body Length, Progeny Production and Lifespan Assay

The body length and number of progenies were observed under a stereomicroscope (Chongqing Optec Instrument Co., Ltd., Chongqing, China). The body length of 30 random worms in each group was measured using microscope graticules. The assay was performed in triplicate. For progeny production, 30 random worms in each group were separately transferred to a new NGM plate with *Escherichia coli* OP50 and grown to the L4 stage. Worms were transferred to fresh NGM plates during the reproductive period, and the remaining eggs were counted on each day. Finally, the total number of eggs laid by worms in the whole life was counted. The assay was performed in triplicate.

For lifespan, 30 worms in each group were transferred daily to fresh NGM plates from L1 stage until death. The number of surviving worms was then counted. Indicators of death included lack of movement, stress movement of the parasite, and lack of pharynx contraction after one or two attempts at gentle touch.

### 3.6. Quantitative Reverse Transcription-PCR

The total RNA of *C. elegans* samples was isolated using the RaPure Total RNA Kit (Magen, Shanghai, China) following the manufacturer’s protocol. In brief, the RLT lysis buffer was used to lyse the worms, and then total RNA was collected and purified using HiPure RNA Mini Columns. Finally, the RNA was dissolved with RNase free water. The purity and concentration of RNA were determined using a NanoDrop spectrophotometer (Thermo Fisher, Waltham, MA, USA). cDNA was synthesized using the HiScript III. All-in-one RT SuperMix Perfect for qPCR Kit (Vazyme, Nanjing, China). Relative quantitative RT-PCR was performed using Rotor-Gene Q (Qiagen, Beijing, China) with a Taq Pro Universal SYBR qPCR Master Mix Kit (Vazyme, Nanjing, China). RT-PCR amplification conditions included denaturation at 95 °C for 10 s, annealing at 55 °C for 30 s, and extension for 32 s at 72 °C. *Act-1* was used for normalization. All primer pairs were synthesized by Synbio Technologies (Jiangsu, China), and the sizes of PCR products were between 100 to 300 bp. The primer sequences used are listed in Table 6. Results were analyzed using the comparative threshold cycle (Ct) method and expressed as the fold change in gene expression (2^−ΔΔCt^).

### 3.7. Statistical Analysis

All results are presented as mean ± SEM. Statistical significance was determined using one-way analysis of variance (ANOVA) followed by Tukey’s multiple-comparison test or Student’s *t*-test (two-tailed) using SPSS version 19.0 (IBM Corporation, New York, NY, USA). Differences were considered statistically significant at *p* < 0.05. 

## 4. Discussion

In this study, we detected the compositions and contents of phenolic compounds in SAP and HCP. In addition, we explored SAP and HCP on the lifespan, body length, and reproductive ability of *C. elegans* subjected to normal, 100 mM sucrose, or 352 mM stearic acid treatments, with the aim of identifying the impact of SAP and HCP on health. Since excessive carbohydrate and fat intake is harmful to health, 100 mM sucrose or 352 mM stearic acid was used to provide excessive nutrients for *C. elegans*, and the protective effects of SAP and HCP on sucrose-incubated and stearic acid-incubated *C. elegans* were investigated.

We detected the seven phenolic compounds of SAP and eight phenolic compounds of HCP to explore the active components of the extracts. Chlorogenic acid, the most phenolic compound presented in SAP in our study, is supposed to account for many of the beneficial anti-aging, antioxidant, anti-diabetic effects observed, as well as against cardiovascular diseases [29,30,31]. In addition, rutin, the main component of HCP, has been reported to exert antioxidant effects by controlling the expression of antioxidant enzymes [32,33]. Chlorogenic acid and rutin may play a key role separately in the protective effect of SAP and HCP in *C. elegans*.

Excessive sugar and lipid treatment severely shortens the lifespan, body length of the *C. elegans* and diminishes its reproductive ability. Treatment with SAP at different concentrations presented different degress of protective effects on normal *C. elegans.* In addition, treatment with 500, 1000, and 2000 μg/mL SAP significantly (*p* < 0.05) prolonged the lifespan, enhanced the reproductive capacity, and increased the body length of sucrose-incubated *C. elegans*. However, treatment with 500 and 1000 μg/mL SAP, except for 2000 μg/mL SAP, presented effective protection of stearic acid-incubated *C. elegans*. This may be related to the different treatment methods used in *C. elegans*. In other words, sugars and lipids may cause varying degrees of destruction in *C. elegans* [28]. Moreover, the positive effect of 500 μg/mL SAP on the reproductive capacity of stearic acid-incubated *C. elegans* was more pronounced than that of 1000 μg/mL SAP. Although 1200 μg/mL HCP shifted the survival curve to the left, there was no significant effect on the average lifespan of normal *C. elegans*. Treatment with 800, 1200 μg/mL HCP, except for 400 μg/mL HCP, had positive effects on the reproductive capacity of normal *C. elegans*. In addition, only 1200 μg/mL HCP significantly (*p* < 0.05) improved the maximum body length of normal *C. elegans*. Interestingly, we found that the effect of HCP on the body length of sucrose-incubated *C. elegans* was not as severe as that of SAP. This may be related to the differences in the phenolic compositions of SAP and HCP. All the three concentrations of HCP effectively improved the lifespan and reproductive capacity of sucrose-incubated *C. elegans*. Similarly, treatment with 400, 800, and 1200 μg/mL HCP significantly (*p* < 0.05) promoted the reproductive capacity of stearic acid-incubated *C. elegans*. However, only 1200 μg/mL HCP had positive effects on the lifespan and body length of stearic acid-incubated *C. elegans*. Overall, our results indicated that SAP and HCP protected *C. elegans* from sugars or lipids. By overall consideration, we used 500 μg/mL SAP and 1200 μg/mL HCP to further investigate the protective mechanism against *C. elegans*.

In *C. elegans*, the major ROS detoxification mechanisms are initiated by the transcription factor *skn-1*, which responds to oxidative stress [34]. Previous studies indicated that *skn-1* promotes the expression of phase II detoxification genes, such as *gst-4* and *gst-7*, which play key roles in increasing oxidative stress resistance and extending lifespan by scavenging free radicals [35]. Zheng et al. [36] reported a significant increase in the lifespan of *C. elegans* via activating the transcription of *skn-1* after supplying chlorogenic acid. This is consistent with our findings that *skn-1* mRNA expression in stearic acid-incubated *C. elegans* was increased with SAP treatment. The TGF-β signaling pathway is important for the regulation of stress responses in organisms [37]. A ligand for the TGF-β signaling pathway, *daf-7*, is related to stress resistance in *C. elegans*. A low expression of *daf-7* indicates a harsh environment and severe stress in *C. elegans* [38]. Our results indicated that 352 mM stearic acid treatment significantly decreased the expression of *daf-7*, suggesting that high stearic acid levels caused damage to *C. elegans*. Treatment with SAP and HCP significantly (*p* < 0.05) increased *daf-7* mRNA expression in stearic acid-incubated *C. elegans*. This was consistent with the phenotypic results that both SAP and HCP showed protective effects against stearic acid-damaged *C. elegans*. In the present study, we found that SAP and HCP treatments had no effects on *skn-1* and *daf-7* mRNA expression in sucrose-incubated *C. elegans*. We only evaluated the gene expression of *C. elegans* at the L4 stage, and further experiments are needed to investigate how SAP and HCP affect *C. elegans* in different environments. As a signaling ligand, *dbl-1* is an important component of the TGF-β signaling pathway; the core components of the *dbl-1* pathway consist of receptors (*daf-4* and *sma-6*) and Smads (*sma-2*, *sma-3*, and *sma-4*) [39]. Loss of function of the signaling ligand, receptors or Smads reduces the postembryonic growth of *C. elegans*, leading to a smaller body size [28]. High sugar and lipid levels disrupt the metabolism of *C. elegans*, thereby causing damage [28]. In our study, 100 mM sucrose and 352 mM stearic acid treatments significantly (*p* < 0.05) reduced the expression of *dbl-1* and *sma-4*, suggesting that high sucrose and stearic acid inhibited the development of *C. elegans*. HCP treatment significantly (*p* < 0.05) improved the expression of *dbl-1* and *sma-4* in sucrose-incubated *C. elegans*. In addition, in normal *C. elegans*, HCP treatment significantly (*p* < 0.05) improved the expression of *dbl-1* compared with that in the SAP group. This was consistent with the phenotypic results showing that HCP treatment increased the maximum body length of normal *C. elegans*. In conclusion, we showed that SAP can effectively extend the lifespan, enhance reproductive ability, and promote growth of *C. elegans* under high stearic acid exposure via *skn-1* and *daf-7*. In addition, we found that HCP can considerably increase the body length of normal *C. elegans* via *dbl-1*, and promote growth of *C. elegans* under stearic acid via *daf-7* and *sma-4*.

In summary, our study demonstrated that SAP and HCP had different phenolic compositions. Additionally, we observed different degrees of protective effects of SAP and HCP treatment on *C. elegans* damaged by high sucrose or high stearic acid levels. Taken together, we speculate that the repair effect of SAP and HCP against damage due to high sucrose or high stearic acid is mainly manifested in the reduction of oxidative stress in *C. elegans*. Collectively, SAP and HCP are expected to ameliorate the damage caused by overnutrition.

## Figures and Tables

**Figure 1 molecules-28-01707-f001:**
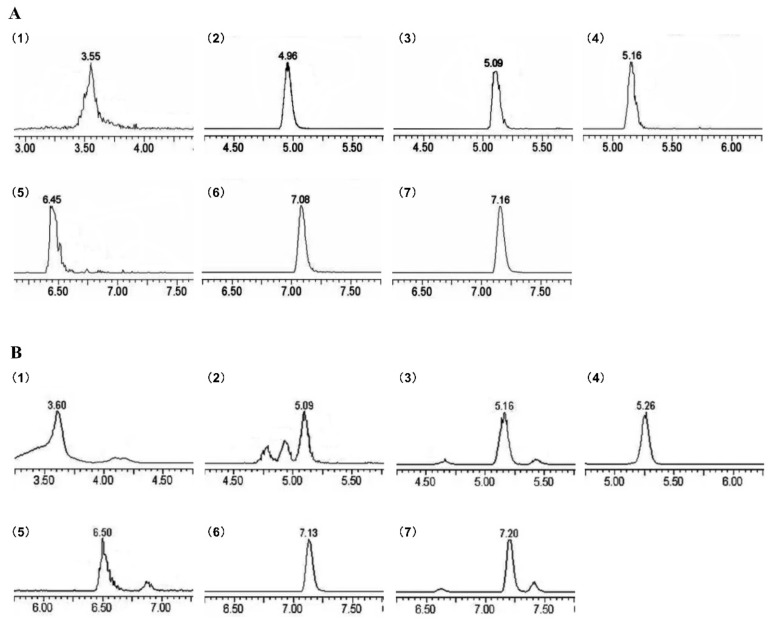
MRM chromatogram of *Sonchus arvensis* Linn. phenolic compounds (SAP) showing the presence of seven phenolic compounds as identified. (**A**) MRM chromatogram of standards; (**B**) MRM chromatogram of SAP. Peaks: (1) chlorogenic acid, (2) scopoletin, (3) rutin, (4) hyperin, (5) quercetin, (6) apigenin, (7) chrysoeriol.

**Figure 2 molecules-28-01707-f002:**
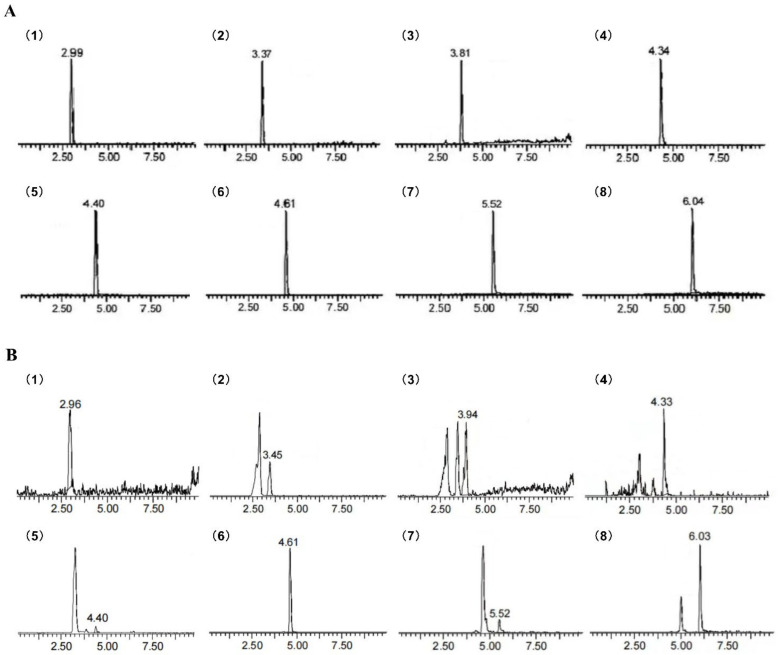
MRM chromatogram of *Hemerocallis citrina* Baroni. phenolic compounds (HCP) showing the presence of eight phenolic compounds as identified. (**A**) MRM chromatogram of standards; (**B**) MRM chromatogram of HCP. Peaks: (1) protocatechin, (2) chlorogenic acid, (3) caffeic acid, (4) taxifolin, (5) gentisic acid, (6) rutin, (7) quercetin, (8) isorhamnetin.

**Figure 3 molecules-28-01707-f003:**
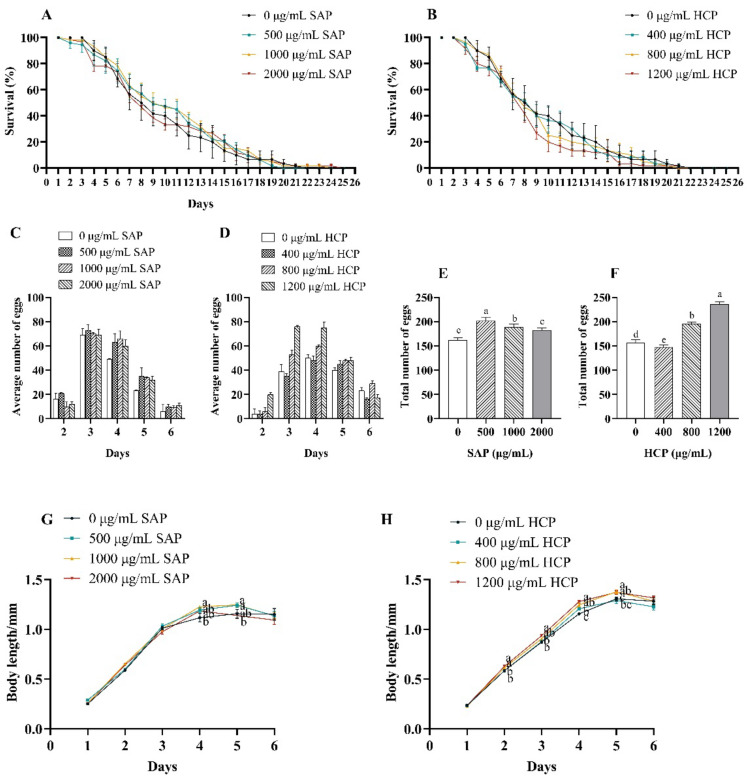
Effects of SAP and HCP on lifespan, reproduction, and body length of the *C. elegans*. (**A**,**B**): effects of SAP (**A**) and HCP (**B**) on the lifespan of *C. elegans*. (**C**,**D**): effects of SAP (**C**) and HCP (**D**) on the number of eggs of *C. elegans* on each day. (**E**,**F**): effects of SAP (**E**) and HCP (**F**) on the total number of eggs of *C. elegans* in the entire spawning period. (**G**,**H**): effects of SAP (**G**) and HCP (**H**) on the body length of *C. elegans*. Data are presented as mean ± SEM (n = 30). Values without a common letter are significantly different at *p* < 0.05.

**Figure 4 molecules-28-01707-f004:**
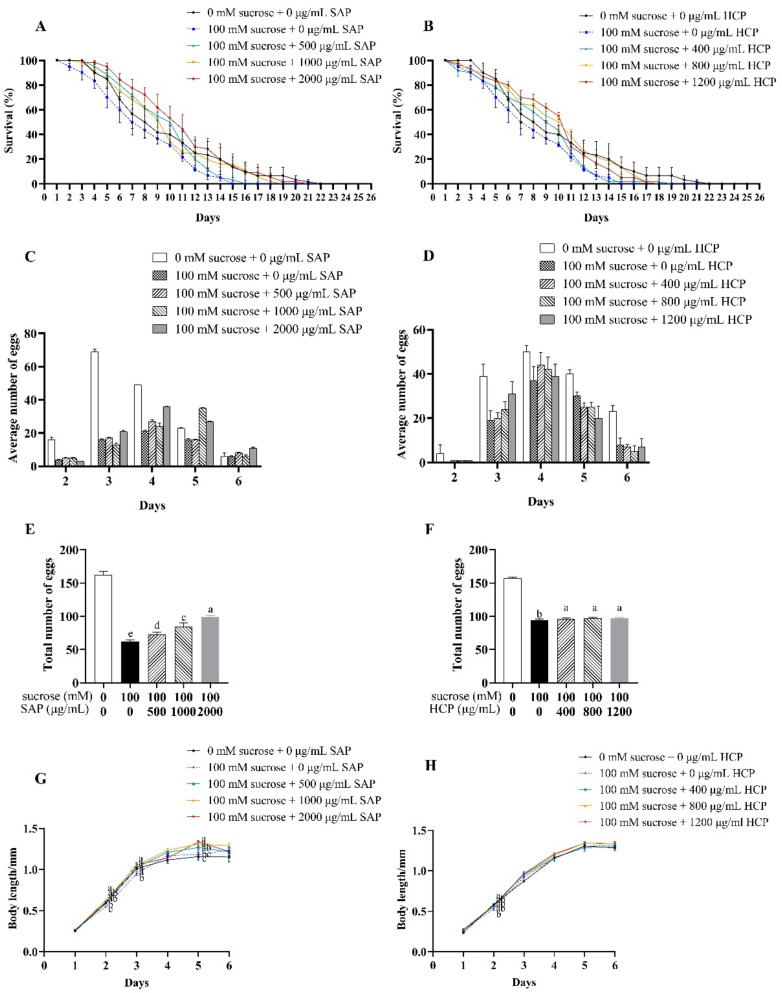
Effects of SAP and HCP on lifespan, reproduction, and body length of the sucrose-incubated *C. elegans*. (**A**,**B**): effects of SAP (**A**) and HCP (**B**) on the lifespan of sucrose-incubated *C. elegans*. (**C**,**D**): effects of SAP (**C**) and HCP (**D**) on the average number of eggs of sucrose-incubated *C. elegans* on each day. (**E**,**F**): effects of SAP (**E**) and HCP (**F**) on the total number of eggs of sucrose-incubated *C. elegans* in the entire spawning period. (**G**,**H**): effects of SAP (**G**) and HCP (**H**) on the body length of sucrose-incubated *C. elegans*. Data are presented as mean ± SEM (n = 30). Values without a common letter are significantly different at *p* < 0.05.

**Figure 5 molecules-28-01707-f005:**
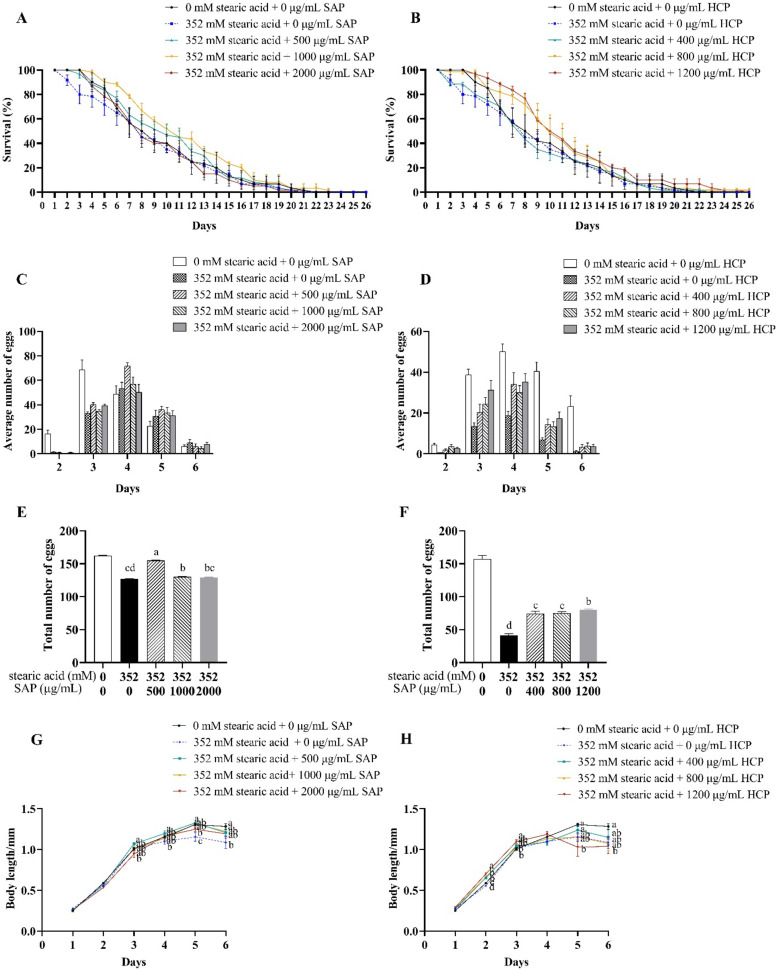
Effects of SAP and HCP on life-span, reproduction, and body length of the stearic acid-incubated *C. elegans*. (**A**,**B**): effects of SAP (**A**) and HCP (**B**) on the lifespan of stearic acid-incubated *C. elegans*. (**C**,**D**): effects of SAP (**C**) and HCP (**D**) on the average number of eggs of stearic acid-incubated *C. elegans* on each day. (**E**,**F**): effects of SAP (**E**) and HCP (**F**) on the total number of eggs of stearic acid-incubated *C. elegans* in the entire spawning period. (**G**,**H**): effects of SAP (**G**) and HCP (**H**) on the body length of stearic acid-incubated *C. elegans*. Data are presented as mean ± SEM (n = 30). Values without a common letter are significantly different at *p* < 0.05.

**Figure 6 molecules-28-01707-f006:**
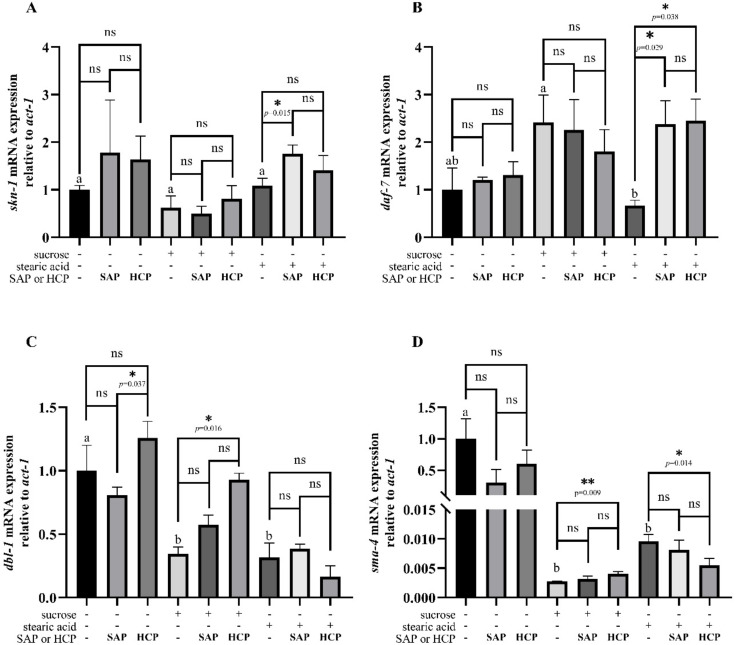
Effects of SAP and HCP on gene expression of *C. elegans*. Worms were treated with SAP (500 μg/mL) or HCP (1200 μg/mL). (**A**) *skn-1* mRNA expressions, (**B**) *daf-7* mRNA expression, (**C**) *dbl-1* mRNA expression, (**D**) *sma-4* mRNA expression. Values are expressed as mean ± SEM. Different superscripts were considered significantly different, ANOVA followed by Tukey’s multiple-comparison test. * *p* <0.05; ** *p* < 0.01; Student’s *t*-test (two-tailed).

**Table 1 molecules-28-01707-t001:** The phenolic compositions of SAP and HCP.

SAP		HCP	
Composition	Content (μg/g Dry Material)	Composition	Content (μg/g Dry Material)
Chlorogenic acid	1061.01 ± 1.12	Rutin	5721.11 ± 10.63
Hyperin	7.87 ± 0.006	Chlorogenic acid	73.62 ± 0.39
Rutin	3.90 ± 0.09	Quercetin	13.02 ± 0.14
Apigenin	1.70 ± 0.012	Caffeic acid	9.35 ± 0.053
Quercetin	0.15 ± 0.04	Protocatechin	3.02 ± 0.02
Scopoletin	0.15 ± 0.0003	Isorhamnetin	1.50 ± 0.12
Chrysoeriol	0.08 ± 0.00003	Taxifolin	0.79 ± 0.003
		Gentisic acid	0.34 ± 0.0008

**Table 2 molecules-28-01707-t002:** The mean lifespan of *C. elegans* treated with SAP and HCP.

SAP (μg/mL)	Average Life (Days)	Relative Life Rate of Change (%)	HCP (μg/mL)	Average Life (Days)	Relative Life Rate of Change (%)
0	8.43 ± 0.18 ^b^		0	8.43 ± 0.13 ^a^	
500	10.38 ± 0.39 ^a^	+23.14	400	9.13 ± 0.63 ^a^	+8.31
1000	10.38 ± 0.81 ^a^	+23.14	800	8.9 ± 0.15 ^a^	+5.64
2000	9.23 ± 0.67 ^ab^	+10.09	1200	8.1 ± 0.75 ^a^	−3.86

Data are presented as mean ± SEM (n = 30). Values without a common letter are significantly different at *p* < 0.05.

**Table 3 molecules-28-01707-t003:** The mean lifespan of sucrose-incubated *C. elegans* treated with SAP and HCP.

Group	Average Life (Days)	Relative Life Rate of Change (%)	Group	Average Life (Days)	Relative Life Rate of Change (%)
Control	8.43 ± 0.13 ^c^		Control	8.43 ± 0.13 ^b^	
Sucrose (100 mM)	7.45 ± 0.00 ^d^	−11.57	Sucrose (100 mM)	7.45 ± 0.00 ^c^	−11.57
Sucrose (100 mM) + SAP (500 μg/mL)	9.60 ± 0.20 ^a^	+28.86	Sucrose (100 mM) + HCP (400 μg/mL)	8.68 ± 0.08 ^b^	+16.44
Sucrose (100 mM) + SAP (1000 μg/mL)	9.13 ± 0.08 ^b^	+22.48	Sucrose (100 mM) + HCP (800 μg/mL)	10.48 ± 0.08 ^a^	+40.60
Sucrose (100 mM) + SAP (2000 μg/mL)	9.80 ± 0.15 ^a^	+31.54	Sucrose (100 mM) + HCP (1200 μg/mL)	10.23 ± 0.23 ^a^	+37.25

Dates are presented as mean ± SEM (n = 30). Values without a common letter are significantly different at *p* < 0.05.

**Table 4 molecules-28-01707-t004:** The mean lifespan of stearic acid-incubated *C. elegans* treated with SAP and HCP.

Group	Average Life (Days)	Relative Life Rate of Change (%)	Group	Average Life (Days)	Relative Life Rate of Change (%)
Control	8.43 ± 0.13 ^bc^		Control	8.43 ± 0.13 ^b^	
Stearic acid (352 mM)	8.25 ± 0.10 ^c^	−2.08	Stearic acid (352 mM)	8.25 ± 0.10 ^b^	−2.08
Stearic acid (352 mM) + SAP (500 μg/mL)	9.88 ± 0.93 ^ab^	+19.70	Stearic acid (352 mM) + HCP (400 μg/mL)	9.38 ± 0.18 ^b^	+13.64
Stearic acid (352 mM) + SAP (1000 μg/mL)	11.08 ± 0.03 ^a^	+34.24	Stearic acid (352 mM) + HCP (800 μg/mL)	12.15 ± 0.45 ^a^	+47.27
Stearic acid (352 mM) + SAP (2000 μg/mL)	9.13 ± 0.03 ^bc^	+10.61	Stearic acid(352 mM) + HCP(1200 μg/mL)	11.08 ± 0.73 ^a^	+34.24

Date are presented as mean ± SEM (n = 30). Values without a common letter are significantly different at *p* < 0.05.

**Table 5 molecules-28-01707-t005:** The effect of relative genes on *C. elegans*.

Gene Name	Function	Ref.
*skn-1*	promote oxidative stress resistance; anti-aging	[22,25]
*daf-7*	enhance stress resistance	[19]
*dbl-1*	the signaling ligand of TGF beta-related pathway; promote growth	[26]
*sma-4*	regulate body size	[27]

**Table 6 molecules-28-01707-t006:** Primer squences.

Genes	Primer Sequences		Product Sizes
	Forward (5′-3′)	Reverse (5′-3′)	
*act-1*	ACTCTGGAGATGGTGTCA	CGTCAGGAAGTTCGTAGG	273
*daf-7*	CGAGAAGAACGAGGATGG	TTGCCTTGACGAAGATACC	152
*skn-1*	ATCCACCAGCATCTCCAT	CTTCTCCATAGCACATCAATC	121
*dbl-1*	TCCGCTTATTGTGTTCAGT	GGTGCCATAATCCAGTCTT	201

## Data Availability

Not applicable.

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
