# Peer review of "Phenolic Compounds from Sonchus arvensis Linn. and Hemerocallis citrina Baroni. Inhibit Sucrose and Stearic Acid Induced Damage in Caenorhabditis elegans"

_molecules, 2023, doi:10.3390/molecules28041707_

Round 1
Reviewer 1 Report
In the manuscript entitled “Polyphenol from Sonchus arvensis Linn and Hemerocallis citrina Baroni Inhibited the Damage of Caenorhabditis elegans Induced by Sucrose and Stearic Acid” the Authors took an effort to determine the protective effect of phenolic rich fractions on selected parameters during in vivo study. The article contains interesting data and is potentially interesting for the readers, but it requires some modifications before being accepted. The answers and comments should be included in new version of manuscript, since they will allow simpler following of presented data.
- First of all please check the proper writing form of “Sonchus arvensis Linn. “ and “Hemerocallis citrina Baroni.” ;
- According to Authors the selected plants are rich sources of polyphenol – and this is misleading since they contain many different phenolic compounds;
- The aim of the study should be presented in the introduction in a better way;
- In the introduction characterize better the chemical constituents of selected plants, as well as present the molecular targets chosen for the study;
- Explain the rationale of usage of selected concentration of sucrose, stearic acid and preparations; explain how concentrations used in the study are correlated to the doses present in human diet; why stearic acid is not presented as mM – please modify this unit!
- Lines 104, 107 – please explain what means that “Each group has three replicates”
- Materials and methods should be presented with more details, since very often they can not be repeated, i.e. mRNA isolation protocol; the RT-PCR procedure; what is the molecular weight of products obtained with designed starters?
- Present the chromatograms from phenolic compounds identification; what means “g” used in the table – row material, dry material or sth else? Did Authors check the phenolics composition and content in original material? Usually, in diet extracts are not used, but the whole plants are present, so did the Authors check the effect of the same dose of the whole extract obtained from plant (raw material) and compare it with the phenolics fraction? What is the correlation of effective dose of phenolics rich fraction with the phenolics content in plant? Instead of usage “*” in Table 2 rather present the comments of analysis within the text. Were there any other compounds present in the preparations rich in phenolic compounds, but not identified by the Authors?
- Please explain in what solvent the SAP and HCP preparations were prepared to obtain the final concentrations for incubation with Caenorhabditis elegans and what was the stock solution concentration?
- During in vivo study was used group treated only with standard treatment (no DMSO)? – please add this data;
- Lines 139-141 – the very low quality of data presentation does not support the conclusion “As shown in Figure 5A-B, the average life span of C. elegans treated with different concentrations of SAP and HCP increased initially and then decreased.”. This sentence should be modified and matched with concentration of preparations; frankly speaking within the manuscript there are other sentences describing in too general/simple way the results – it needs to be obligatory modified! What is more, in 3.2 chapter there is almost no analysis of results describing the effects of both preparations on survival of Caenorhabditis elegans.
- All figures have to be enlarged!
- There is no explanation why different concentrations of preparations (SAP and HCP) were used in experiments, not the same – it will allow to compare the effects quantitively and suggest the selection of more effective preparation.
- Line 144 – Authors use term ”two kinds of polyphenols”, which is not correct, since they worked with preparations containing mixtures of isolated different types of phenolic compounds. Within the whole manuscript they used wrong terms describing as “polyphenol” the phenolic rich preparation or phenolic rich fraction - even in the title. It needs to be corrected within the whole manuscript!
- I do not see the need to present percentage data with two decimals “24.28%, 16.57%, and 12.47%”, please correct this.
- The title of Table 4 is misleading. Instead of “The mean life span of sucrose-damaged C. elegans treated with SAP and HCP” try to emphasize the effects of preparations on C. elegans co-incubated with sucrose. Instead of “model” present “Sucrose 100 mM”. Within the manuscript also is used term “sucrose-damaged C. elegans” – please, change this expression according to the suggestion – “incubated, treated with sucrose”; there was used term “high concentration of sucrose ‘ -there is no comment about using in the experiment other concentrations of sucrose, as well as explanation bout choosing the 100 mM; similar comments are for Table 5 illustrating the effect of stearic acid and the chapter 3.4;
- Figure 2 B- correct the legend since sucrose was added; pleas, carefully check all the figures and legends – there are many mistakes present! 2C-D the number of days is different – correct this; 2EF – explain if the total number of eggs was counted after 6 days? In Figure 2 there is many times stearic acid presented, whereas the chapter 3.3 describes sucrose effect – please comment this’
- Please explain the rationale for studying the effects of the highest concentration of preparation, when they negatively effected survival of C. elegans for at least 11 days;
- Chapter 3.5 – the Authors often use word “significantly” – it should be rather written “statistically significant”; there is very weak analysis describing in a quantitative way the most relevant results – please, improve this; please explain why “Worms were treated with SAP (500 μg/mL) or HCP (1200 μg/mL)” – especially when previous results demonstrated better effects of HCP 800 μg/mL; did the Authors checked the effect on protein level?
- Line 349- chlorogenic acid is not a polyphenol but phenolic acid;
- Discussion should be enriched and presents more detailed information about potential protective activities of studied plants.
The Authors should present the explanations within the manuscript. I suggest the major revision.
Author Response
Response to Reviewer 1 Comments
Point 1: First of all please check the proper writing form of “Sonchus arvensis Linn.” and “Hemerocallis citrina Baroni.”
Response 1: Thank you for reminding. We checked and the writing form of “Sonchus arvensis Linn.” and “Hemerocallis citrina Baroni.” are determined correctly.
Point 2: According to Authors the selected plants are rich sources of polyphenol – and this is misleading since they contain many different phenolic compounds.
Response 2: Thank you for your advice. We have corrected this in Introduction: “The extracts of Sonchus arvensis Linn. scavenge the free radicals, which may be due to the presence of functional components include rutin, quercetin, catechin, and myricetin.” (line 50-52); “The strong antioxidant activity of Hemerocallis citrina Baroni. can be attributed to the action of antioxidant compounds such as phenolic compounds.” (line 62-63).
Point 3: The aim of the study should be presented in the introduction in a better way.
Response 3: Thank you for your advice. We have added the aim of study in the Introduction: “In the present study, to better understand the bioactivity of SAP and HCP, … The results will help investigate the mechanism of SAP and HCP and define health claims related to the consumption of Sonchus arvensis Linn. and Hemerocallis citrina Baroni.” (line 83-90)
Point 4: In the introduction characterize better the chemical constituents of selected plants, as well as present the molecular targets chosen for the study.
Response 4: Thanks for the advice. We changed the description of chemical constituents (line 50-53; line57-60) and added the description of molecular targets (line 74-82) in the introduction.
Point 5: Explain the rationale of usage of selected concentration of sucrose, stearic acid and preparations; explain how concentrations used in the study are correlated to the doses present in human diet; why stearic acid is not presented as mM – please modify this unit!
Response 5: Thanks for the suggestions.
According to the established model, we used 100 mM sucrose and 352 mM stearic acid. We add citations in the chapter 2.4. Before formal experiments, we set up several doses of SAP and HCP and selected the appropriate concentration range;
100 mM sucrose is 3.42 g sucrose in 100 mL water, which is less than half the sugar of a regular sugary drink in human diet. The recommended intake of Hemerocallis citrina Baroni. is about 20-30 g, and the recommended intake of Sonchus arvensis Linn. is about 150 g in human diet. The concentrations used in the study is lower than the doses present in human diet;
We changed “100 μg/mL stearic acid” into “352 mM stearic acid”. We are very grateful to this suggestion. The reason why we used different measurement unit is that we referred to different literatures and followed the measurement methods used in different laboratories.
Point 6: Lines 104, 107 – please explain what means that “Each group has three replicates”.
Response 6: Thanks for reminding us. We have changed “Each group has three replicates” to “The assay was performed in triplicate”. (line 142, 147)
Point 7: Materials and methods should be presented with more details, since very often they can not be repeated, i.e. mRNA isolation protocol; the RT-PCR procedure; what is the molecular weight of products obtained with designed starters?
Response 7: Thanks for your advice. The mRNA isolation protocol and the RT-PCR procedure have been presented in the chapter 2.6. The size of PCR products are between 100 to 300 bp. We added Product Sizes in Table 1.
Point 8: Present the chromatograms from phenolic compounds identification; what means “g” used in the table – row material, dry material or sth else? Did Authors check the phenolics composition and content in original material? Usually, in diet extracts are not used, but the whole plants are present, so did the Authors check the effect of the same dose of the whole extract obtained from plant (raw material) and compare it with the phenolics fraction? What is the correlation of effective dose of phenolics rich fraction with the phenolics content in plant? Instead of usage “*” in Table 2 rather present the comments of analysis within the text. Were there any other compounds present in the preparations rich in phenolic compounds, but not identified by the Authors?
Response 8: Thank you for your suggestions.
The chromatograms from phenolic compounds identification have been presented (Figure 1 and 2);
“g” used in the table means dry material;
The phenolics composition and content were presented in Table 2. And the data presented the content of phenolic compounds in the dry original material;
We plan to detect the effects of the extracts from plants and compare with several phenolic components. At present, it is under preparation.
We did not check the effect of the phenolics rich fraction, but we plan to do that in future. Therefore, it is difficult to tell the effective dose of exact phenolic fraction.
The “*” in Table 2 has been deleted.
There are some phenolic compounds that we haven’t tested, and a note was added to the chapter 3.1 (line 178-183).
Point 9: Please explain in what solvent the SAP and HCP preparations were prepared to obtain the final concentrations for incubation with Caenorhabditis elegans and what was the stock solution concentration?
Response 9: Thank you for your comments. In the chapter 2.4: “SAP or HCP completely dissolved in DMSO and configured to 2 mg/mL stock solution. For assays, SAP (0, 500, 1000 and 2000 μg/mL) or HCP (0, 400, 800 and 1200 μg/mL) were added to Escherichia coli OP50 at a rate of 2%.” (line 131-136)
Point 10: During in vivo study was used group treated only with standard treatment (no DMSO)? – please add this data.
Response 10: Thanks for your advice. We added the description to the chapter 2.4: “SAP or HCP completely dissolved in DMSO and configured to 2 mg/mL stock solution. For assays, SAP (0, 500, 1000, and 2000 μg/mL) or HCP (0, 400, 800, and 1200 μg/mL) were added to Escherichia coli OP50 at a rate of 2%.” (line 131-136)
Point 11: Lines 139-141 – the very low quality of data presentation does not support the conclusion “As shown in Figure 5A-B, the average life span of C. elegans treated with different concentrations of SAP and HCP increased initially and then decreased.”. This sentence should be modified and matched with concentration of preparations; frankly speaking within the manuscript there are other sentences describing in too general/simple way the results – it needs to be obligatory modified! What is more, in 3.2 chapter there is almost no analysis of results describing the effects of both preparations on survival of Caenorhabditis elegans.
Response 11: Thank you for your suggestions.
In the chapter 3.2, we changed this sentence into “As shown in Table3 and Figure 3A-B, treatment with 500-1000 μg/mL SAP …Treatment with HCP at concentrations ranging from 400 to 1200 μg/mL had no significant effect on the average life span of C. elegans.”;
We modified the descriptions in Results;
In the chapter 3.2, we added the analysis of results describing the effects of SAP and HCP on the survival of C. elegans. (line 204-212)
Point 12: All figures have to be enlarged!
Response 12: Thanks for the advice. We enlarged all figures.
Point 13: There is no explanation why different concentrations of preparations (SAP and HCP) were used in experiments, not the same – it will allow to compare the effects quantitively and suggest the selection of more effective preparation.
Response 13: Thank you for your advice. Based on the results of the preliminary experiment that a higher concentration of HCP caused damage to C. elegans, we used different concentrations of SAP and HCP. We added a brief explanation in the chapter 3.2 (line 202-204).
Point 14: Line 144 – Authors use term ”two kinds of polyphenols”, which is not correct, since they worked with preparations containing mixtures of isolated different types of phenolic compounds. Within the whole manuscript they used wrong terms describing as “polyphenol” the phenolic rich preparation or phenolic rich fraction - even in the title. It needs to be corrected within the whole manuscript!
Response 14: Thanks for your suggestions. We changed “two kinds of polyphenols” into “two kinds of extrats”. We changed “polyphenol” into “phenolic compounds”.
Point 15: I do not see the need to present percentage data with two decimals “24.28%, 16.57%, and 12.47%”, please correct this.
Response 15: Thank you for your advice. We have corrected this. (line 221, 226)
Point 16: The title of Table 4 is misleading. Instead of “The mean life span of sucrose-damaged C. elegans treated with SAP and HCP” try to emphasize the effects of preparations on C. elegans co-incubated with sucrose. Instead of “model” present “Sucrose 100 mM”. Within the manuscript also is used term “sucrose-damaged C. elegans” – please, change this expression according to the suggestion – “incubated, treated with sucrose”; there was used term “high concentration of sucrose ‘-there is no comment about using in the experiment other concentrations of sucrose, as well as explanation bout choosing the 100 mM; similar comments are for Table 5 illustrating the effect of stearic acid and the chapter 3.4;
Response 16: Thanks for your suggestions.
The model in Table 4 has been changed to “Sucrose (100 mM)”. The expression has been changed to “sucrose-incubated C. elegans”;
We changed “high concentration of sucrose” to “100 mM sucrose”, and added citation in the chapter 2.4;
The model in Table has been changed to “Stearic acid (100 μg/L)”. The expression has been changed to “stearic acid-incubated C. elegans”.
Point 17: Figure 2 B- correct the legend since sucrose was added; pleas, carefully check all the figures and legends – there are many mistakes present! 2C-D the number of days is different – correct this; 2EF – explain if the total number of eggs was counted after 6 days? In Figure 2 there is many times stearic acid presented, whereas the chapter 3.3 describes sucrose effect – please comment this’
Response 17: Thank you for your advice.
We checked all the figures and corrected the legend;
We corrected the number of days of Figure 2C-D;
We added note to the Figure: “the total number of eggs of C. elegans in the entire spawning period.” And we added explanation in the chapter 2.5 (line 144-147);
Stearic acid should not be presented in Figure 2. We have corrected it.
Point 18: Please explain the rationale for studying the effects of the highest concentration of preparation, when they negatively effected survival of C. elegans for at least 11 days;
Response 18: Thanks for the suggestion. We added the explanation in Discussion (line 432-463).
Point 19: Chapter 3.5 – the Authors often use word “significantly” – it should be rather written “statistically significant”; there is very weak analysis describing in a quantitative way the most relevant results – please, improve this; please explain why “Worms were treated with SAP (500 μg/mL) or HCP (1200 μg/mL)” – especially when previous results demonstrated better effects of HCP 800 μg/mL; did the Authors checked the effect on protein level?
Response 19: Thanks for your suggestions.
We quoted relevant p-values at all mentions of “significantly”;
We admit that Western-Blot may be more efficient way than Real-time PCR to explored the mechanism of repair effects. If we check the effect on protein level, we need a greater number of worms to detect each protein, it is more difficult to operate;
In Discussion: “Treatment with HCP at 800, 1200 μg/mL significantly enhanced the productive capacity of normal C. elegans. Especially, 1200 μg/mL HCP significantly improved the maximum body length of normal C. elegans. … Treatment with HCP at 400, 800, 1200 μg/mL significantly promoted the productive capacity of stearic acid-incubated C. elegans. Moreover, 1200 μg/mL HCP also had significant positive effect on the life span and body length of stearic acid-incubated C. elegans.”
We did not check the effect on protein level. Because we didn’t know the exact mechanism at first. RT-PCR can help us screen for more related genes.
Point 20: Line 349- chlorogenic acid is not a polyphenol but phenolic acid;
Response 20: Thank you for your advice. “Chlorogenic acid, the most polyphenols…” has been changed into “Chlorogenic acid, the most phenolic compounds…”. (line 425-426)
Point 21: Discussion should be enriched and presents more detailed information about potential protective activities of studied plants.
Response 21: Thank you for your advice. We revised the discussion.
Reviewer 2 Report
In this study, authors reported the effects of SAP and HCP on the life span, reproductive ability and growth of wild C. elegans and two damaged C. elegans model. It showed that both SAP and HCP have protective effects against C. elegans damage caused by high sucrose or high stearic acid, and Gst-10 and daf-7 mediated oxidative stress resistance may be involved in the protective effects . The result of this study can lay a certain foundation for us to further understand the physiological activity of natural products. The data of the paper is sufficient to the conclusion, but there are still some minor problems that need to be fixed.
1. Line 70-72
In the “Materials”, it should be clearly explained what part of the plant were used as the experimental materials, the leaves? Flowers? Fruit? Or root?
2. Line 139, 146
In the second paragraph of 3.2, “Figure 5” appears many times. Is there something wrong? Are these results should be marked Figure 1?
3. Line 139-140
“As shown in Figure 5A-B, the aver-age life spanof C. eleganstreated with different concentrations of SAP and HCP increased initially and then decreased. ” This description of the results contradicts the life span trend in Figure 1B. Please check it.
4. Figure 1 (E,F) effects of SAP and HCP on the number of eggs of C. eleganson each day. The experimental methods for these results were not clearly introduced in the “Methods” section. And the author described “the number of progenywere observed......” in the “Methods” section, but only “the total number of eggs laid by C. elegans......” in the results, if the two are the same meaning? Please describe it clearly in the method.
5. Line 177-178; Line 222-223
“ Effects of SAP and HCP on the Body Length, Progeny Production and Life Span of Sucrose-damaged C. Elegans” and “Effects of SAP and HCP on the Body Length, Progeny Production and Life Span of Stearic Acid-Damaged C. Elegans” When did “Sucrose” or “Stearic Acid” add in the culture? And then, how to add SAP or HCP in the culture? All of these compounds added together with OP50? The methods of these experiments have not be described for detail in the “Method”.
6. Line 275-282
This comment may be appear in the “Discussion”, Please reconsider it.
7. In the second paragraph of “Discussion”section, there were only the re-description of the experimental results, if the experimental results can be discussed in combination with the existing literature reports, it may arouse readers' interest more.
8. “The major polyphenol found in SAP and HCP was chlorogenic acid and rutin, respectively.” (Line 131)
Weather the “Effects of SAP and HCP on the Body Length, Progeny Production and Life Span of C. Elegans” is related to the major polyphenol found in SAP or HCP? Why did the authors do the component analysis ? Please discuss it in the “Discussion”.
9. Western-Blot may be more efficient way than Real-time PCR to explored the mechanism of repair effects.
Author Response
Point 1: Line 70-72: In the “Materials”, it should be clearly explained what part of the plant were used as the experimental materials, the leaves? Flowers? Fruit? Or root?
Response 1: Thank you for your advice. We added this part in the “Materials”: “The leaves of Sonchus arvensis Linn. were collected … The buds of Hemerocallis citrina Baroni. were collected.” (line 100-101)
Point 2: Line 139, 146: In the second paragraph of 3.2, “Figure 5” appears many times. Is there something wrong? Are these results should be marked Figure 1?
Response 2: Thanks for reminding us. After modification, the results in the chapter 3.2 are marked Figure 3 (line 213).
Point 3: Line 139-140: “As shown in Figure 5A-B, the average life span of C. elegans treated with different concentrations of SAP and HCP increased initially and then decreased. ” This description of the results contradicts the life span trend in Figure 1B. Please check it.
Response 3: Thanks for your advice. We corrected it. “As shown in Figure 5A-B, …” has been changed into “As shown in Figure 3A and Table 3, the survival curve of C. elegans treated with 500 and 1000 μg/mL SAP were obviously shifted to the right. In addition, treatment with 500 and 1000 μg/mL SAP significantly (p < 0.05) prolonged the average life span of C. elegans. As shown in Figure 3B and Table 3, the survival curve of 1200 μg/mL HCP was shifted to the left. Treatment with HCP at concentrations ranging from 400 to 1200 μg/mL had no significant effect on the average life span of C. elegans.” (line 204-211)
Point 4: Figure 1 (E,F) effects of SAP and HCP on the number of eggs of C. elegans on each day. The experimental methods for these results were not clearly introduced in the “Methods” section. And the author described “the number of progeny were observed......” in the “Methods” section, but only “the total number of eggs laid by C. elegans......” in the results, if the two are the same meaning? Please describe it clearly in the method.
Response 4: Thanks for the suggestions. Worms were transferred to fresh NGM plates during the reproduction period, and the eggs left were counted. “The average number of eggs” means “The number of eggs on the old medium was counted on each day”. “The total number of eggs” means “the total number of eggs laid by worms in the whole life was counted”. We added the description into Methods (line 144-147).
Point 5: Line 177-178; Line 222-223: “ Effects of SAP and HCP on the Body Length, Progeny Production and Life Span of Sucrose-damaged C. Elegans” and “Effects of SAP and HCP on the Body Length, Progeny Production and Life Span of Stearic Acid-Damaged C. Elegans” When did “Sucrose” or “Stearic Acid” add in the culture? And then, how to add SAP or HCP in the culture? All of these compounds added together with OP50? The methods of these experiments have not be described for detail in the “Method”.
Response 5: Thank you for your advice. We added the descriptions in the Methods: “Sucrose plates were prepared by adding sucrose (100 mM, sterile filtered) into NGM. Stearic acid plates were prepared by adding stearic acid (352 mM, sterile filtered) into NGM. SAP or HCP completely dissolved in DMSO and configured to 2 mg/mL stock solution. For assays, SAP (0, 500, 1000 and 2000 μg/mL) or HCP (0, 400, 800 and 1200 μg/mL) were added to Escherichia coli OP50 at a rate of 2%.” (line 131-135)
Point 6: Line 275-282: This comment may be appear in the “Discussion”, Please reconsider it.
Response 6: Thanks for the advice. We deleted this comment in the results. (line 362-371)
Point 7: In the second paragraph of “Discussion” section, there were only the re-description of the experimental results, if the experimental results can be discussed in combination with the existing literature reports, it may arouse readers' interest more.
Response 7: Thank you for your advice. The “Discussion” has been enriched. (line 431-463)
Point 8: “The major polyphenol found in SAP and HCP was chlorogenic acid and rutin, respectively.” (Line 131). Weather the “Effects of SAP and HCP on the Body Length, Progeny Production and Life Span of C. Elegans” is related to the major polyphenol found in SAP or HCP? Why did the authors do the component analysis ? Please discuss it in the “Discussion”.
Response 8: Thank you for your advice. We added this part in the “Discussion”. (line 424-430)
Point 9: Western-Blot may be more efficient way than Real-time PCR to explored the mechanism of repair effects.
Response 9: Thank you for your suggestion. We admit that Western-Blot is a more efficient. If we check the effect on protein level, we need a greater number of worms to detect each protein. It is more difficult to operate. Because we didn’t know the exact mechanism at first. RT-PCR can help us screen for more related genes.
Round 2
Reviewer 1 Report
The authors answered most of my concerns, however I suggest to modify Figure 1 and Figure 2, and only enlarged chromatograms of studied preparations with identified peaks (named as numbers representing each compound). After this modification I recommend manuscript acceptance.
